# Comparison of shape quantification methods for genomic prediction, and genome-wide association study of sorghum seed morphology

Lisa Sakamoto[1,2], Hiromi Kajiya-Kanegae[3], Koji Noshita[4,5], Hideki Takanashi[1], Masaaki Kobayashi[6], Toru Kudo[6], Kentaro Yano[6], Tsuyoshi Tokunaga[7], Nobuhiro Tsutsumi[1], Hiroyoshi Iwata[1] *

1 Graduate School of Agricultural and Life Sciences, University of Tokyo, Tokyo, Japan, 2 JSPS Research Fellow, Tokyo, Japan, 3 Research Center for Agricultural Information Technology, NARO, Ibaraki, Japan, 4 Department of Biology, Kyushu University, Fukuoka, Japan, 5 PRESTO, JST, Saitama, Japan, 6 Faculty of Agriculture, Meiji University, Kanagawa, Japan, 7 EARTHNOTE Co., Ltd., Okinawa, Japan

* aiwata@mail.ecc.u-tokyo.ac.jp

**Data Availability Statement:** All seed counter shape data are available from the https://github.

## Abstract

Seed shape is an important agronomic trait with continuous variation among genotypes. Therefore, the quantitative evaluation of this variation is highly important. Among geometric morphometrics methods, elliptic Fourier analysis and semi-landmark analysis are often used for the quantification of biological shape variations. Elliptic Fourier analysis is an approximation method to treat contours as a waveform. Semi-landmark analysis is a method of superimposed points in which the differences of multiple contour positions are minimized. However, no detailed comparison of these methods has been undertaken. Moreover, these shape descriptors vary when the scale and direction of the contour and the starting point of the contour trace change. Thus, these methods should be compared with respect to the standardization of the scale and direction of the contour and the starting point of the contour trace. In the present study, we evaluated seed shape variations in a sorghum (*Sorghum bicolor* Moench) germplasm collection to analyze the association between shape variations and genome-wide single-nucleotide polymorphisms by genomic prediction (GP) and genome-wide association studies (GWAS). In our analysis, we used all possible combinations of three shape description methods and eight standardization procedures for the scale and direction of the contour as well as the starting point of the contour trace; these combinations were compared in terms of GP accuracy and the GWAS results. We compared the shape description methods (elliptic Fourier descriptors and the coordinates of superposed pseudo-landmark points) and found that principal component analysis of their quantitative descriptors yielded similar results. Different scaling and direction standardization procedures caused differences in the principal component scores, average shape, and the results of GP and GWAS.

com/risasakamoto/Comparison-of-shape-quantification-methods.

**Funding:** This research was supported by grant (JPMJCR12B5) from the JST/CREST. The first author, L.S., was supported by grant (17J05431) from the JSPS.EARTHNOTE Co. contributed to the preparation of materials. PRESTO provided support in the form of salaries for authors KN, but did not have any additional role in the study design, data collection and analysis, decision to publish, or preparation of the manuscript. The specific roles of these authors are articulated in the 'author contributions' section.

**Competing interests:** This research was supported by grant (JPMJCR12B5) from the JST/CREST. The first author, L.S., was supported by grant (17J05431) from the JSPS.EARTHNOTE Co. contributed to the preparation of materials. PRESTO provided support in the form of salaries for authors KN, but did not have any additional role in the study design, data collection and analysis, decision to publish, or preparation of the manuscript. This does not alter our adherence to PLOS ONE policies on sharing data and materials.

## Introduction

Seed shape is an important agronomic trait of cereal crops because it is directly/indirectly related to the quality as well as the volume of grain production [1–5]. Genetic dissection of seed shape is important for efficient genetic improvement. Because seed shape has continuous variations among genotypes, its quantification is indispensable for applying statistical genetics methods to these variations. The ratio between one-dimensional measurements of a seed, e.g., its length, width, and thickness, is a simple but frequently used index in statistical genetics methods, such as quantitative trait locus mapping [4,6]. However, this index is not sufficient to describe shape diversity [7]. Geometric morphometrics, which is a comprehensive shape description method, can provide a more descriptive quantitative index for seed shape, and has been successfully used in related quantitative genetics studies [3,8,9].

Cultivated sorghum (*Sorghum bicolor* Moench) ranks fifth in the world cereal grains production and is a versatile source of food as well as biomass for animal feed and biofuel. Because sorghum germplasm collections exhibit diverse seed shapes, it is difficult to describe these variations using simple shape indices, such as the ratios of seed length, width, and thickness. Although the shape and size of sorghum seeds are both important breeding targets, only the genetic dissection of seed weight variations has been studied [10–14].

Among geometric morphometrics methods, elliptic Fourier analysis [15] and semi-landmark analysis [16] are commonly used to describe biological contour shape variations. Elliptic Fourier analysis is an approximation method by Fourier in which a contour is treated as a waveform. Semi-landmark analysis is a method of superimposed points in which the differences of multiple contour positions are minimized. However, no detailed comparison of these methods has been undertaken. Moreover, the shape descriptors used in these methods vary with the variation of the scale and direction of the contour and the starting point of the contour trace. Thus, the methods should be compared under various scale and direction standardization procedures for the contour and the starting point of the contour trace. In elliptic Fourier analysis, standardization is often performed based on the first harmonic ellipsoid [17]. In semi-landmark analysis, standardization is often performed by generalized Procrustes analysis [18]. It should be noted that it may be difficult to evaluate the potential of the methods accurately based on any specific combination of shape description methods and contour standardization procedures.

In the present study, we analyzed seed shape variations in a sorghum germplasm collection to model the association between the shape variations and genome-wide single-nucleotide polymorphisms (SNPs) through genomic prediction (GP) [17] and genome-wide association studies (GWAS) [18]. In the analysis, we used all possible combinations between three shape description methods and eight standardization procedures for the scale and direction of the contour and the starting point of the contour trace. These combinations were compared in terms of GP accuracy and the GWAS results. We also compared simple shape indices, i.e., the ratios of seed length, width, and thickness, with the above-mentioned geometric morphometrics methods to demonstrate the superiority of the latter.

## Materials and methods

### Plant materials and their genotyping

In this study, we used 329 accessions of a sorghum germplasm collection (S1 Table). The marker genotype dataset was a subset of 423 sorghum accessions that included *Sorghum bicolor* and *Sorghum x almum*. The accessions were genotyped for 127,587 SNPs by using double-digest RAD-Seq [19]. A RAD-Seq library was generated by BglII and MseI digestion of

genomic DNA, and was sequenced with 100-bp single-end reads on an Illumina HiSeq2000 sequencer (Illumina, Inc., San Diego, CA, USA). Preprocessing for the adapter trimming and quality filtering of the sequence data was conducted as described by Ohyanagi et al. [20]. Pre-processed reads were then aligned to the reference genome (Sbicolor_v2.1, http://www.phytozome.net/) by using the Burrows–Wheeler Aligner with the default options [21]. SNPs were genotyped by using Stacks [22]. A reference panel of 65 inbred lines with whole-genome sequence data was used to impute the missing marker genotypes of the remaining 404 inbred lines for the 127,587 SNPs. The genome sequences of two lines were newly determined by the authors and those of the other 63 inbred lines by Kobayashi et al. [23], Mace et al. [24], and Zheng et al. [25]. BEAGLE ver. 3.3.254 [26] was used to impute the missing SNP genotypes. The nucleotide sequence data obtained in this study are archived in the DDBJ Sequenced Read Archive under accession number PRJDB4150. Of all the SNPs, 39,904, which had minor allele frequency (MAF) > 0.03 in the 329 accessions, were used in this study. A PCA plot of the accessions by the SNPs and neighbor-joining trees was built using the ape package in R [27], as shown in S1 Fig.

## Acquisition and analysis of seed images

We took an image of 10 seeds for each accession using a flatbed scanner (GT-X820, EPSON, Japan). The resolution of the image was 6,400 dpi with 32-bit colors. Seeds were placed on a sticky mending tape posted on a 35-mm slide mount. Profile images of seeds were taken from the front and the side views of the seeds (Fig 1).

Using MATLAB Image Processing Toolbox (MATLAB_R2015a), we performed the following image analysis tasks. First, we created a binary image by setting non-seed regions as the background and seed regions as the foreground. Then, we applied erosion and dilation filters to the binary image to fill holes and remove noise. Finally, we determined a base landmark manually (Fig 1) at the position where the seed was attached to a parental plant, obtained the seed contour coordinates starting from the base landmark, and recorded them as Freeman chain codes of eight directions.

## Quantitative evaluation of seed size and shape

Seed area, length, width, and thickness were defined and measured as follows. Area was defined as the number of pixels segmented as a seed, because all the images were taken on the same scale. Length was determined as the longest distance from the base landmark to all points on the contour (Fig 1). Width and thickness were determined as a range of values, i.e., the difference between maximum and minimum values, on vertical axes (i.e., axes vertical to the length direction) in the front and the side views, respectively (Fig 1). We calculated the length-width ratio and the length-thickness ratio as simple shape indices.

The quantitative evaluation of seed shape was performed in five steps: (1) sampling of contour coordinates from the starting point of the contour trace, (2) standardization of contour coordinates for the direction and scale of the contour and the starting point of the contour trace, (3) quantitative shape description of the standardized contour coordinates, (4) summarization of variations in the descriptors by principal component analysis (PCA), and (5) estimation of the average of shape descriptors for each accession. Before the quantitative evaluation of seed shape, we sampled 500 equally spaced points from a contour starting from the base landmark (including the base landmark). Moreover, we used four contour standardization procedures for the direction and scale of the contour and the starting point of the contour trace. The procedures are based on simple landmarks (SL), generalized Procrustes analysis (GPA), the first ellipsoid with the starting point set semi-major axis (FESA), and on the first

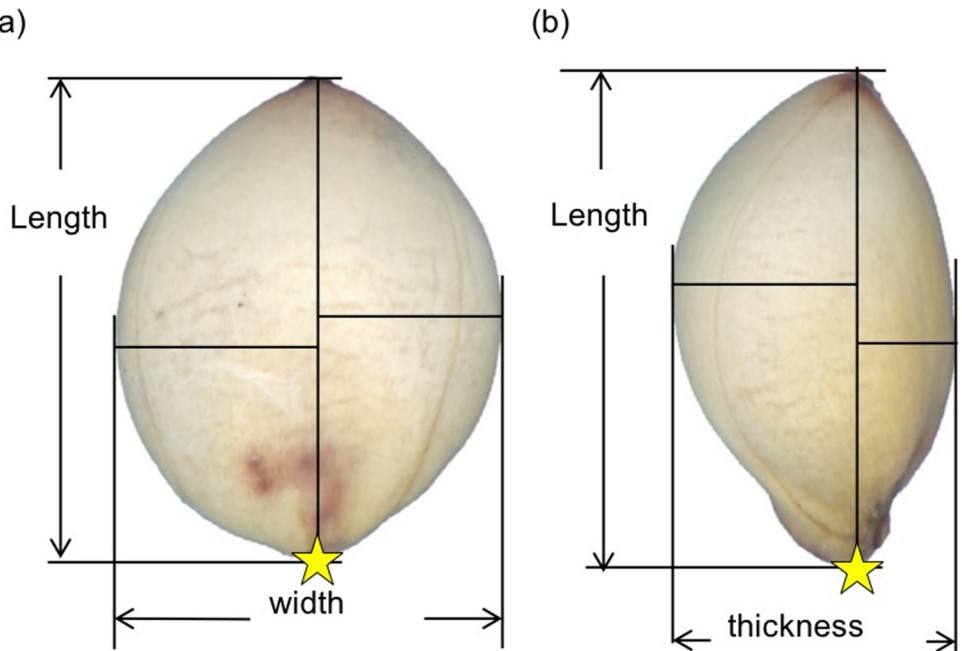

**Fig 1. Front (a) and side (b) views of sorghum seed.** Yellow stars represent the base landmark taken manually at the point where the seed attached to a parental plant.

ellipsoid with the starting point set base landmark (FESL). We used the statistical analysis software R (ver. 3.4.2) [28] for all the procedures.

SL was performed based on two landmarks: the base landmark and the centroid of the contour coordinates. A counter was scaled and rotated so that its base landmark and centroid were located at $(c, 0)$ and $(0, 0)$, respectively. Scale standardization was realized by setting $c = 1$. GPA is a well-known landmark-based procedure, in which a fixed number of points on the contours are scaled and rotated so that the total sum of the Euclidean distances between the points is minimized among all contours [29]. In the present study, GPA was performed over the 500 equally spaced points, which include the base landmark on the contour. We used the function gpa of the R package "shapes" [30] for GPA. Scale standardization was then performed with the scale option of gpa. FESA, which was proposed by Kuhl and Giardina [15], is a procedure frequently used in elliptic Fourier analysis. A counter was scaled and rotated so that the terminal points of the semi-major axis of the first ellipsoid became $(-c, 0)$ and $(c, 0)$. Thereby, the starting point was moved from the base landmark to the terminal points of the semi-major axis of the first ellipsoid. Scale standardization was then realized by setting $c = 1$. We used the function "ef2nef" in the R script "Shape on R" [31]. FESL was performed as in the case of FESA, but the starting point was fixed at the base landmark in FESL. Then, scale standardization was realized by setting $c = 1$.

In the quantitative description of the standardized contours, two methods, i.e., a method based on elliptic Fourier descriptors (EFD) and a method based on the coordinates of superposed pseudo-landmark points (SPP), were applied by using R. EFD quantifies the shape and size variations in contour coordinates by Fourier series expansions [15]. The coefficients of the Fourier expansions are used by EFD as shape descriptors. In this study, we quantified the counter coordinates of a seed in two resolutions: one using the first 20 harmonics with $20 \times 4$ coefficients and one using the first 250 harmonics with $250 \times 4$ coefficients of EFD using the function "cood2ef" in "Shape on R" [31]. The first 20 harmonics captured major variations;

they missed some variations in high-frequency levels and acted as a low-pass filter. The first 250 harmonics did not miss any variations (i.e., converted all variations in the coordinates into Fourier coefficients). SPP quantifies the shape and size of contour coordinates by using the coordinates of superposed pseudo-landmark points (SPP) [16] directly as coefficients. As described above, we sampled 500 equally spaced points from a contour starting from its base landmark and superposed them with the standardized of contour coordinates for the direction and scale of the contour and the starting point of the contour trace. We directly used the coordinates of the 500 points as SPP.

Variations in the coefficients of EFD or SPP were summarized by principal component analysis (PCA) performed based on the variance-covariance matrix of individual seed samples (10 seeds per accession). The average seed shape of each accession was obtained as the average of the shape descriptors over 10 seeds.

## Reconstruction of seed contours

Shape variations explained by PCA, the average seed shape of each accession, and the seed shape predicted in GP were visualized by contour reconstruction. To examine the morphological meaning of each principal component (PC), we inversely calculated the values of EFD or SPP descriptors corresponding to the mean, $-2$ SD and $+2$ SD scores of each PC based on the eigenvector matrix [32]. Contour coordinates were obtained by inverse Fourier transformation from the EFD descriptors (i.e., coefficients) using the function "ef2cood" in "Shape on R" [31].

## Phenotypic correlation and cluster analysis

Phenotypic correlations are calculated as Pearson's product moment correlation between morphological characteristics. In the cluster analysis, the distance between morphological characteristics was calculated as $d = 1 - r^2$, where $r$ is a correlation coefficient between the characteristics. The analysis was performed using the centroid method.

## GP and GWAS

Genotypic values of all traits including the PC scores of the EFD or SPP descriptors were predicted based on genome-wide markers. As geometric morphometrics traits, we used the first 30 PCs of the EFD or SPP descriptors for the prediction. A prediction model was constructed for each PC separately. In the prediction, we used GBLUP and reproducing kernel Hilbert spaces (RKHS) [33]. GBLUP is linear regression based on the additive relational matrix, whereas RKHS is non-linear regression based on a Gaussian kernel. We used the function "BGLR" in the related R package for the regression. In RKHS, we used a multi-kernel model with $h = X/M$, where X is {1/5, 1, 5} and M is the median squared Euclidean distance between lines, following the guidelines in the BGLR package documentation [33]. After predicting the scores of the first 30 PCs, we calculated the EFD or SPP descriptors from the predicted PC scores as described in Iwata et al. [8]. The predicted counter was reconstructed from the descriptors as described above. Prediction accuracy was evaluated by 10-fold cross validation. As described in Iwata et al. [9], the accuracy of the contour reconstruction was calculated by

$$Q^2 = 1 - \sum_{l=1}^{L} \sum_{t=1}^{T} \{[\overline{x_{lt}} - \widehat{x_{lt}}]^2 + [\overline{y_{lt}} - \widehat{y_{lt}}]^2\} / \sum_{l=1}^{L} \sum_{t=1}^{T} \{[\overline{x_{lt}} - \overline{\overline{x_t}}]^2 + [\overline{y_{lt}} - \overline{\overline{y_t}}]^2\}, \quad (1)$$

where $t$ is the sequential index of contour points, $\widehat{x_{lt}}$ and $\widehat{x_{lt}}$ are the predicted coordinates of $l$-th accession at the $t$-th point, $\overline{x_{lt}}$ and $\overline{y_{lt}}$ are the average coordinates of $l$-th accession at the $t$-th point, and $\overline{\overline{x_t}}$ and $\overline{\overline{y_t}}$ are the means of $\overline{x_{lt}}$ and $\overline{y_{lt}}$, respectively, over all accessions. Because we sampled 500 equally spaced points, we set the value of $T$ to 500 for calculating $Q^2$. Higher $Q^2$

represents better prediction accuracy. To evaluate the variation attributable to random split samples in the 10-fold cross-validation, we repeated the 10-fold cross-validation 10 times on different split samples. In each replication, we used an identical random split sample for all methods to enable paired comparison of the prediction accuracy. A difference among methods and procedures was tested using ANOVA analysis in R.

GWAS was performed for the first four PCs of geometric morphometrics traits. Using the function "GWAS" in the R package "rrBULP"[34], we performed GWAS with a mixed model [18]. To reduce the influence of the confounding effect of population stratification, we used a model considering population structure (Q) and family relationships (K). Q was calculated from the PCA of marker genotypes scores. The number of PCs for Q was set to 6. K was generated from the marker genotype by using the "A.mat" function of the R package "rrBULP." Determining significant SNPs was set to 5% FDR threshold by the Benjamini–Hochberg method [35].

## Results

### Principal components of descriptors

PCA extracts similar shape variations among the quantification methods (Fig 2). Phenotypic correlations in SPP and EFD with two different resolution levels (i.e., the first 20 harmonics and the first 250 harmonics) were more than 0.99 or less than −0.99 (i.e., the absolute value of the correlations was more than 0.99). Because the PCA for both methods yielded similar results, we will describe only the results for SPP hereafter.

Shape variations explained by PCs were different among the direction standardization procedures and the scaling procedures (Fig 3). PC1 of FESL explains the variations caused by seed orientation. When scaling was not applied, PC1 of SL, GPA, and FESA explained seed size, and PC2 of SL, GPA, and FESA explained the length-width ratio and the length-thickness ratio. When scaling was applied, PC1 of SL, GPA, and FESA explained the length-width ratio and length-thickness ratio. The remaining PCs of SL, GPA, and FESA explained different types of shape variations from the PCs mentioned above.

Cluster analysis was performed to evaluate the similarity among the size, the simple shape index, and the first to fourth PC scores of the descriptors. The combinations of the methods and the procedures were roughly divided into five clusters (Fig 4). The *n*-th principal component scores of SL, GPA, and FESA with scaling belonged to the same cluster as their (*n* + 1)-th principal component scores without scaling. The first cluster was related to seed length and thickness, whereas the second cluster was related to the length ratio. The third to fifth clusters represented other features.

In all procedures, the cumulative contribution of PCs was larger than 90% with the first four components, and larger than 99.3% with the first thirty components. Thus, the first four components explained a sufficient portion of the shape and size variation to carry out the subsequent genetic analysis.

### Average contour shape

The average contour shapes of all accessions were visualized by the inverse transformation (Fig 5). FESA yielded more rounded shapes than GPA, SL, and FESL (Fig 5A) in "IS14290", "IS15945", "HANGETSUTOSUI", "NYIRARUMOGO", and "THIMBA RED". By comparing the individual 10 seeds and the average shape (Fig 5B), we see that GPA, FESL, and SL reproduced the indentation, but FESA did not.

**Fig 2. Seed shape variations explained by PCs of each quantification method.** The first four components of FESA for the front view when scaling was applied. Coefficients calculated from eigenvectors at −2 SD (orange), +2 SD (blue), and mean (black) for each PC.

### Genomic prediction of contour shape

The prediction accuracy $Q^2$ of the reconstructed contour shape is shown in S2 Table. The $Q^2$ statistic for scaling was significantly different ($p < 0.05$). The accuracy was better when scale was not standardized in all cases (Fig 6). The $Q^2$ statistic for direction standardization was significantly different ($p < 0.05$). The best $Q^2$ statistics were different between the front view and the side view. FESL exhibited lower accuracy than GPA, SL, and FESA for the front view, but higher for the side view.

### GWAS results

Significant SNPs were different among the quantification methods (Tables 1 and S3). No significant SNP was detected in the simple shape and size indices, i.e., front-area, side-area, length, width, thickness, length-width ratio, and length-thickness ratio (Fig 7). The scaling and directional standardization procedures caused differences in the set of significant SNPs. However, we were unable to identify the most suitable combination of quantification methods and standardizing procedures for GWAS. All SNPs detected in GWAS with a p value are shown in S3 Table.

Four traits exhibited significant association at Chr04.59049748. Traits belonging to the same length thickness ratio cluster as these four traits had −log(p) values greater than 2 at Chr04.59049748 (Table 2). This suggests that all traits belonging to that length thickness ratio cluster had significant association at Chr04.59049748 at the 1% probability level.

### Discussion

The choice of quantification method, i.e., EFD or SPP, did not affect the results (Fig 2). This has already been pointed out by Rohlf [36]. EFD and SPP yield identical contour distances because distance is invariant under orthogonal rotations of a vector space, and the Fourier

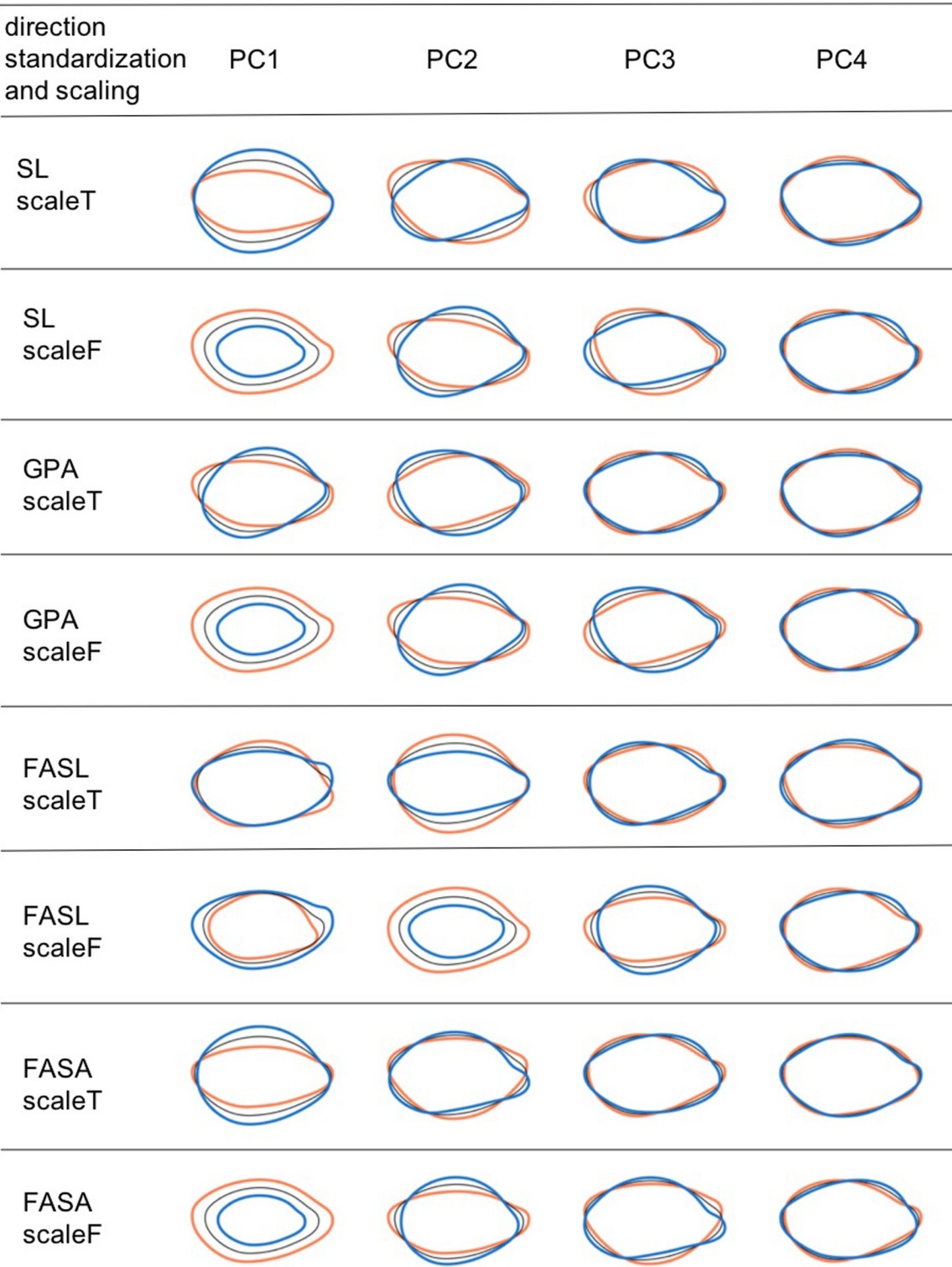

**Fig 3. Seed shape variations explained by PCs of each standardization procedure.** The first four components of SPP for the side view. Coefficients calculated from eigenvectors at −2 SD (orange), +2 SD (blue), and mean (black) for each PC. Eight scale and direction standardization procedures are shown. ScaleT implies that scaling was applied. ScaleF implies that scaling was not applied.

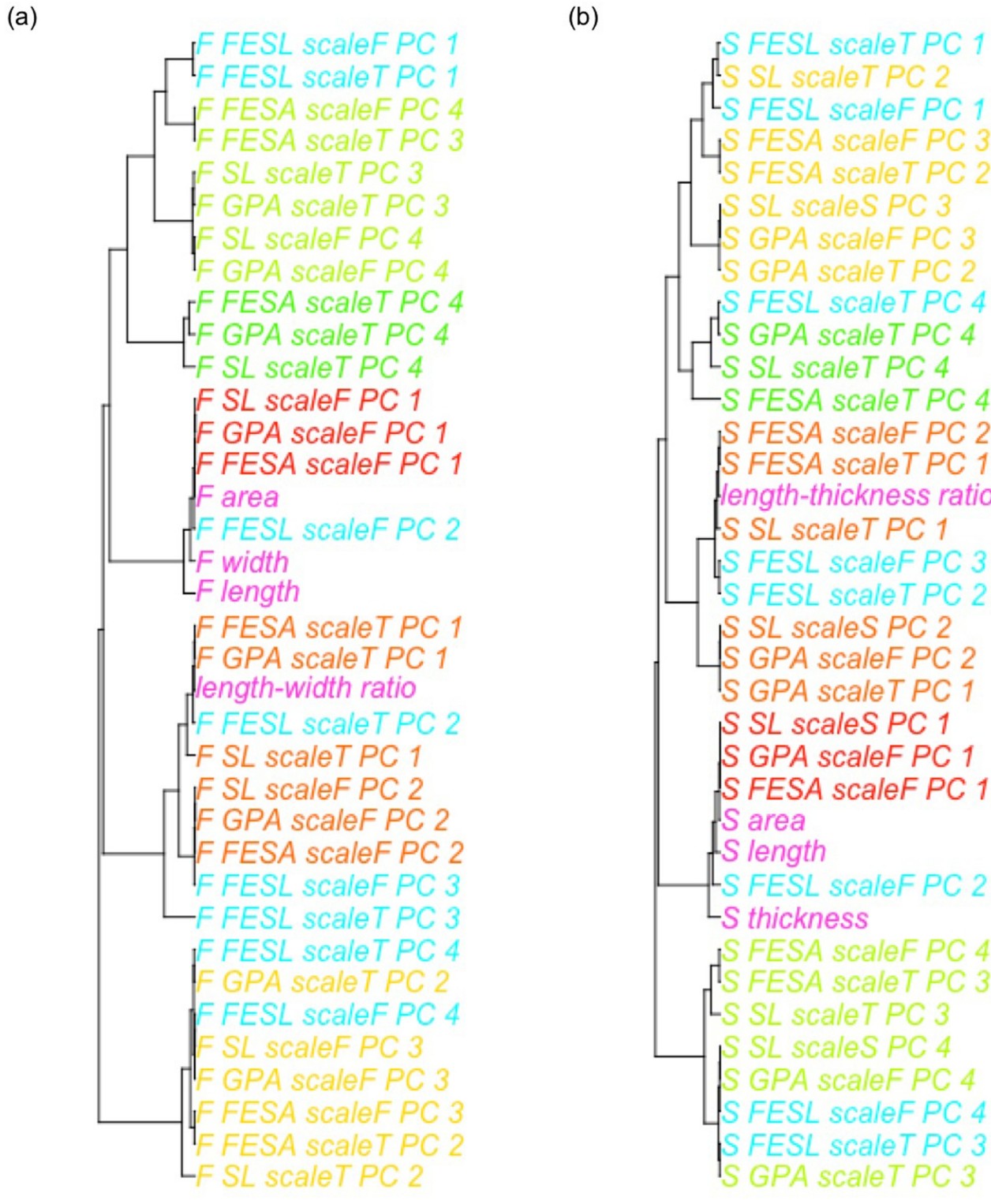

**Fig 4. Cluster analysis of size, simple shape indices, and the scores of the first to fourth PC.** (a) Shape variations for the front view (b) Shape variations for the side view. The combinations of quantification methods and procedures are expressed as [direction of view (front or side)]_ [direction standardization procedure]_[scaling procedure]_[order of PC]. The size and simple shape indices are in pink, FESL are in sky blue, and the *n*-th PC scores of SL, GPA, and FESA with scaling as well as their (*n*+1)-th PC scores without scaling are in other colors.

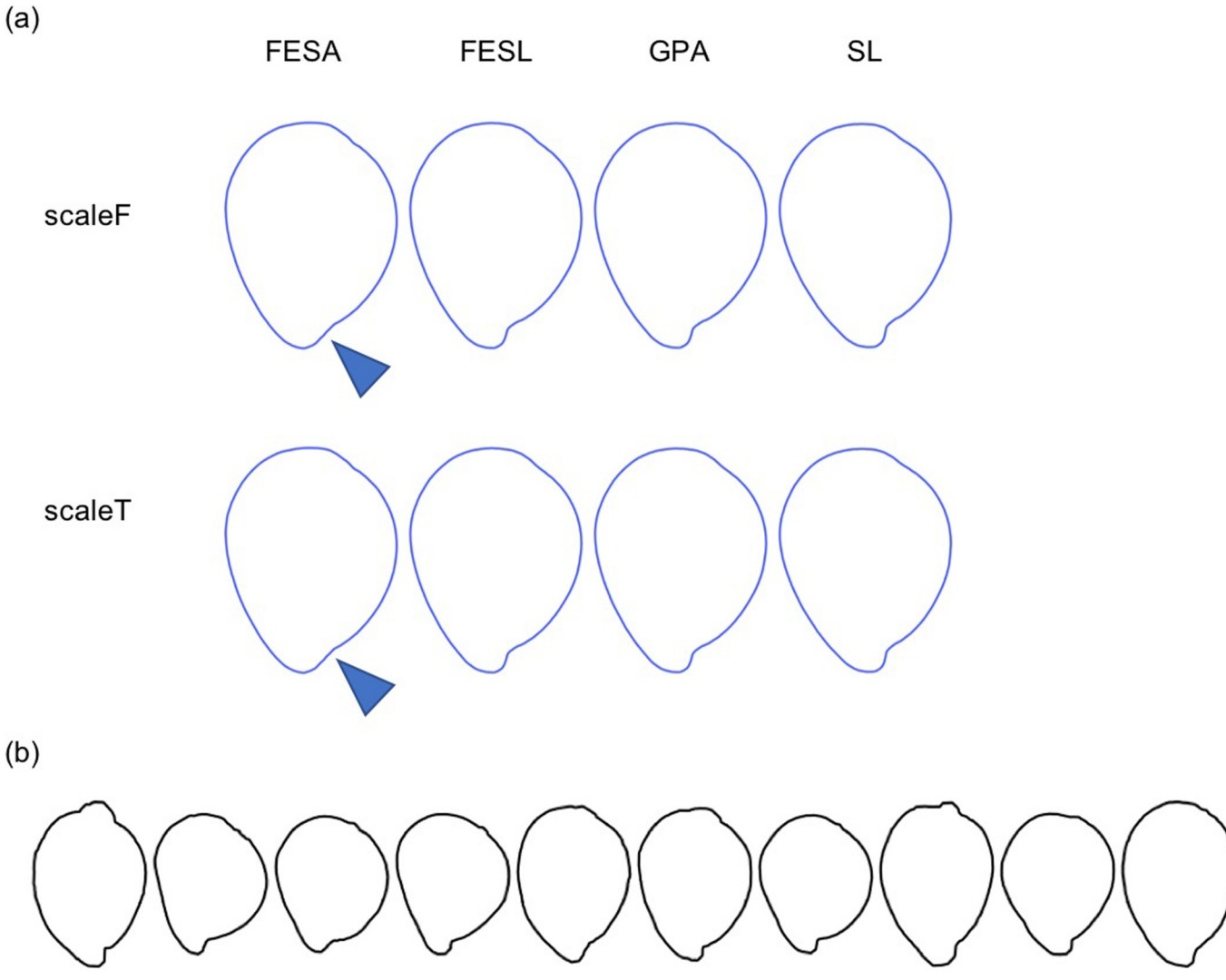

**Fig 5. Average contour shapes.** (a) Blue lines represent the average shape of "NYIRARUMOGO". Columns correspond to direction standardization procedures; triangles represent repaired indentations. (b) Black lines represent 10 individual "NYIRARUMOGO" seeds.

transform with a sufficient number of harmonics is a congruent transformation. Thus, analyses depending only on distances (e.g., principal component analysis, linear discriminant analysis) yield the same results for both EFD and SPP. Based on phenotypic correlation analysis, EFD with 200 harmonics and EFD with 20 harmonics yielded nearly the same results. This suggests that EFD with more than 20 harmonics mainly captures "noise" rather than "signal" in sorghum seeds. That is, EFD can reduce the dimensionality of the data by concentrating "signal" on harmonics of low dimensionality. This point is an advantage of EFD over SPP, which has dimensionality $2 \times n$, where $n$ is the number of sampling points.

According to cluster analysis, some geometric morphometrics clusters represented features other than length and aspect ratio. Thus, length and aspect ratio are insufficient for capturing the whole variations in sorghum seed shape. Furthermore, according to PCA, FESL behaves differently form the other direction standardization procedures. In SL, GPA, and FESA, the $n$-th PC scores when scaling was applied and the $(n+1)$-th principal component scores without

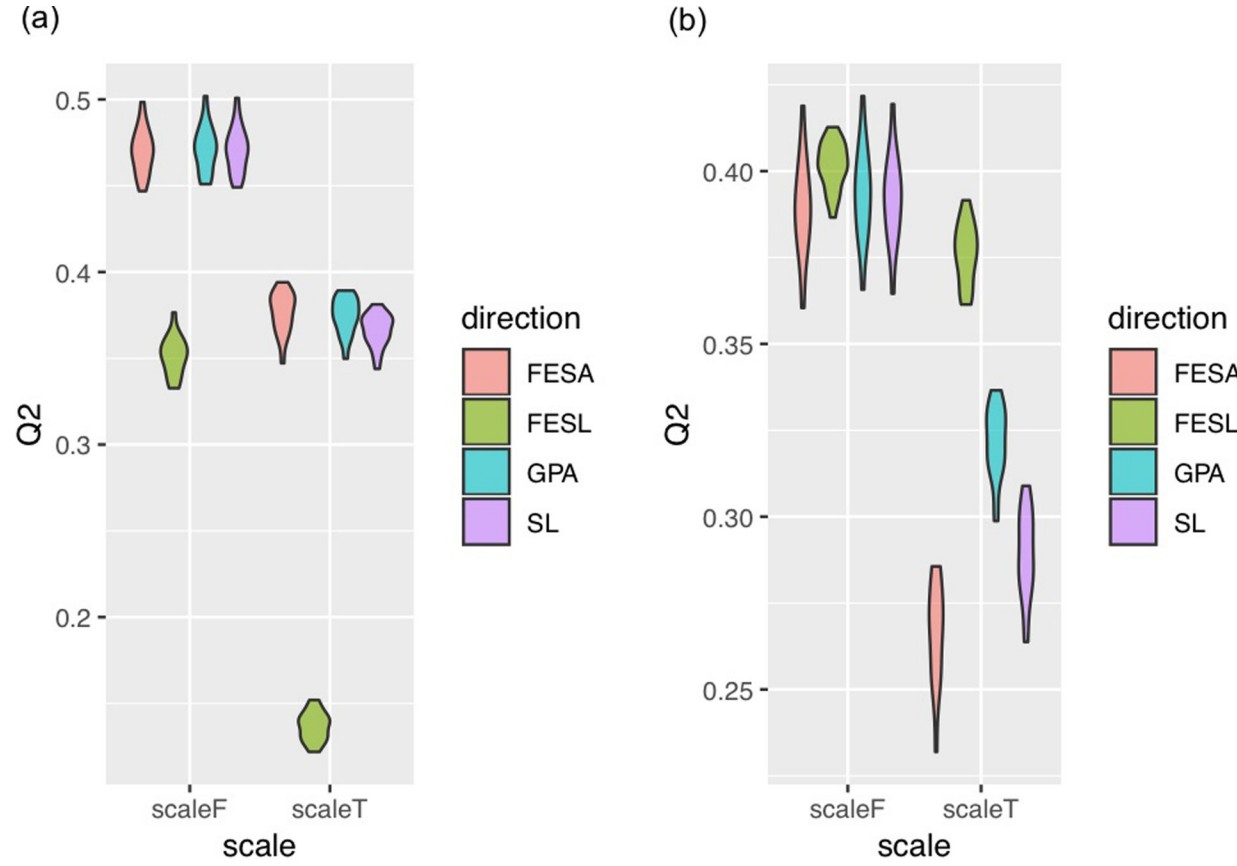

**Fig 6. Prediction accuracy $Q^2$ of seed shape.** Each violin plot corresponds to a direction and scale standardization procedure, and represents the range of $Q^2$ values for seed shape prediction for the front view (a) and the side view (b) in the 10 replications of the 10-fold cross-validation.

scaling had similar characteristics. However, as they were not identical, it can be argued that scaling and direction standardization affect genetic analysis.

Even if the contours of the same seeds were quantified, the average seed shape of each accession was different among direction standardization procedures. In FESA, the average seed

**Table 1. SNPs detected in GWAS.**

| Chr | Pos | Trait |
|---|---|---|
| 1 | 50413644 | S_FESL_scaleT_PC1 |
| 4 | 59021202:59049748 | S_FESL_scaleT_PC1 |
| | | S_GPA_scaleT_PC1 |
| | | S_GPA_scaleF_PC2 |
| | | S_SLM_scaleF_PC2 |
| 5 | 9112888 | S_GPA_scaleT_PC1 |
| | | S_GPA_scaleF_PC2 |
| | | S_SLM_scaleF_PC2 |
| 7 | 11836986 | F_SLM_scaleT_PC1 |
| 9 | 5503744 | F_FESL_scaleT_PC3 |
| 10 | 6445901 | F_FESA_scaleT_PC4 |

SNPs detected in GWAS with 5% FDR threshold; Chr: chromosome number; Pos: position

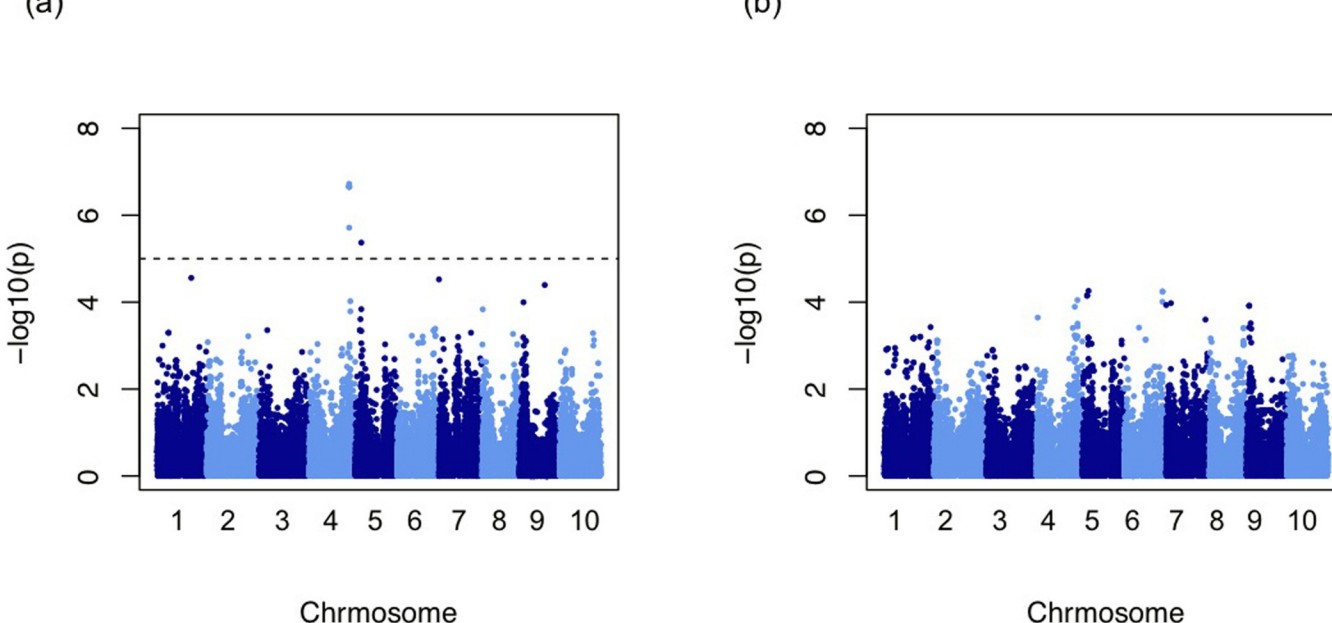

**Fig 7. Manhattan plot of GWAS results.** The broken line shows the 5% FDR threshold. (a) Scores of the second PC when GPA was applied in the side direction. (b) Length-thickness ratio.

shape, particularly around the base landmark, was different form that in SL, GPA, and FESL. Unlike SL, GPA, and FESL, FESA did not use information of the base landmark. Thereby, biological homology among seeds is not completely ensured, particularly around the indentation, and shape features were drowned out by averaging. This indicates that the representative contour of a genotype depends on its orientation. Furthermore, the averaging process may miss varietal features of genotypes, and thus feature analysis may become difficult.

The $Q^2$ statistics of different scaling and direction standardization procedures were largely different, suggesting that scaling and direction standardization influence the accuracy of GP. It should be noted that both the predicted values and the observed values were influenced because accuracy was calculated based on the average shape. Prediction accuracy was higher when scaling was not applied. This may suggest that the size of sorghum seeds is easier to predict than their shape. Prediction accuracy was different among the direction standardization procedures when scaling was performed. The most accurate direction standardization procedures were different between the front and the side views, suggesting that the best procedure may depend on contour shape.

**Table 2. −Log(p) values at Chr04.59049748 in GWAS.**

| Trait | -log(p) value |
| --- | --- |
| S_FESA_scaleF_PC_2 | 3.73 |
| S_FESA_scaleT_PC_1 | 3.62 |
| S_FESL_scaleF_PC_3 | 2.35 |
| S_GPA_scaleF_PC_2 | 6.72 |
| S_GPA_scaleT_PC_1 | 6.70 |
| S_SLM_scaleF_PC_2 | 6.77 |
| S_SLM_scaleT_PC_1 | 3.43 |
| length-thickness ratio | 3.21 |

No significant SNP was detected in the simple shape indices, i.e., the ratios of seed length, width, and thickness. By contrast, significant SNPs were detected in shape characteristics quantified by geometric morphometric approaches, suggesting the advantage of these approaches in quantitative genetics analysis of biological shape. Scaling and direction standardization influenced the result of GWAS (Table 1); however, we were unable to determine the best procedure because the scaling and the directional standardization procedures yielded different GWAS results, and we did not know which result was true. In GWAS, all traits related to the length-thickness ratio (i.e., traits in the length thickness ratio cluster) exhibited significant associations at the 1% probability level at Chr04.59049748. Those traits, however, did not have −log(p) values larger than the FDR 5% threshold owing to the correction for multiple testing. In this case, however, these traits seem to be related to true functional polymorphisms.

On chromosome 4, in the region from 59,021,202 to 59,049,748 bp, there are Sobic.004G 247000 which is homologs of maize *Gln-4* gene [37] related to seed weight in maize, and Sobic.004G245000 which is syntenic genes of Arabidopsis *AHK*4 gene [38, 12] related to seed weight in Arabidopsis. These genes have been pointed out by the sorghum seed mass obtained by QTL analysis, GWAS, and genome analysis [11–14]. These genes may be related to the significant association demonstrated in this study.

In conclusion, the choice of quantification method for sorghum seed shape did not affect shape analysis; the direction and scaling standardization procedures, however, did affect the results.

## Supporting information

**S1 Table. Sorghum accessions used in this study.**
(XLSX)

**S2 Table. Prediction accuracy of the reconstructed contour shape, $Q^2$.**
(CSV)

**S3 Table. All SNPs detected in GWAS with a p value.**
(XLSX)

**S1 Fig.** Neighbor-joining trees (a) and PCA plot of these accessions (b).
(TIFF)

## Author Contributions

**Conceptualization:** Kentaro Yano, Hiroyoshi Iwata.

**Data curation:** Hiromi Kajiya-Kanegae, Koji Noshita, Hideki Takanashi, Masaaki Kobayashi, Toru Kudo, Kentaro Yano, Nobuhiro Tsutsumi.

**Formal analysis:** Lisa Sakamoto, Hiromi Kajiya-Kanegae.

**Funding acquisition:** Tsuyoshi Tokunaga, Nobuhiro Tsutsumi.

**Methodology:** Lisa Sakamoto.

**Project administration:** Lisa Sakamoto, Hiroyoshi Iwata.

**Resources:** Tsuyoshi Tokunaga.

**Supervision:** Hiroyoshi Iwata.

**Writing – original draft:** Lisa Sakamoto, Hiroyoshi Iwata.

**Writing – review & editing:** Lisa Sakamoto, Hiromi Kajiya-Kanegae, Koji Noshita, Hideki Takanashi, Masaaki Kobayashi, Toru Kudo, Kentaro Yano, Nobuhiro Tsutsumi, Hiroyoshi Iwata.

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
