## [Decision Letter · Decision Letter 0]

28 Aug 2019

PONE-D-19-15031

Comparison of shape quantification methods for genomic prediction, and genome-wide association study of sorghum seed morphology

PLOS ONE

Dear Dr. Iwata,

Thank you for submitting your manuscript to PLOS ONE. After careful consideration, we feel that it has merit but does not fully meet PLOS ONE’s publication criteria as it currently stands. Therefore, we invite you to submit a revised version of the manuscript that addresses the points raised during the review process.

We would appreciate receiving your revised manuscript by Oct 12 2019 11:59PM. To enhance the reproducibility of your results, we recommend that if applicable you deposit your laboratory protocols in protocols.io, where a protocol can be assigned its own identifier (DOI) such that it can be cited independently in the future. For instructions see: http://journals.plos.org/plosone/s/submission-guidelines#loc-laboratory-protocols

We look forward to receiving your revised manuscript.

Kind regards,

Sujan Mamidi, Ph.D.

Academic Editor

PLOS ONE

Journal Requirements:

2. Our internal editors have looked over your manuscript and determined that it is within the scope of our Future Crops Call for Papers. This collection of papers is headed by a team of Guest Editors for PLOS ONE. The Collection will encompass a diverse range of research articles on enhanced agronomic production, guaranteeing food security and neglected crop species.  Additional information can be found on our announcement page: https://collections.plos.org/s/future-crops.

If you would like your manuscript to be considered for this collection, please let us know in your cover letter and we will ensure that your paper is treated as if you were responding to this call. If you would prefer to remove your manuscript from collection consideration, please specify this in the cover letter.

This research was supported by grant (JPMJCR12B5) from the JST/CREST. The first author, L.S., was supported by grant (17J05431) from the JSPS.

We note that one or more of the authors are employed by a commercial company:EARTHNOTE Co., Ltd. and/or PRESTO.

6. Please ensure that you refer to Figure 4 in your text as, if accepted, production will need this reference to link the reader to the figure.

Reviewers' comments:

Reviewer's Responses to Questions

**Comments to the Author**

1. Is the manuscript technically sound, and do the data support the conclusions?

Reviewer #1: Partly

Reviewer #2: Yes

2. Has the statistical analysis been performed appropriately and rigorously? 

Reviewer #1: Yes

Reviewer #2: Yes

3. Have the authors made all data underlying the findings in their manuscript fully available?

Reviewer #1: No

Reviewer #2: No

4. Is the manuscript presented in an intelligible fashion and written in standard English?

Reviewer #1: No

Reviewer #2: Yes

5. Review Comments to the Author

Reviewer #1: This MS with goal to compare different seed shape quantification methods and identify the genomic loci associated with seed shape. Authors found the scaling and the direction standardization procedures would cause different PC scores, average shape, GWAS and GP results. Before published, there are some points authors need to clarification and correct.

In the abstract, authors highlighted the methods, it would be better to give more background, and while the detailed comparison is important in introduction.

Is the genome sequencing data (RAD-Seq data) generated in the present study? If yes, I think authors need to provide more detailer results. If not, please cite where the data from. Because I did not find any germplasm information in the MS.

In row 158, 168 and 190, for "function ef2cood in shape on R", authors cited two different papers. need to be corrected.

In row 225, authors used rrBLUP to run GWAS, the paper need to be cited. And authors used Q and K in the GWAS, while not mention where the Q and K from.

In the table 1, authors need to show more information, such as P value for the association.

In rpw 375, authors wrote "The best procedure for the genetic analysis of biological shape should be chosen based on the shape features", can authors point out which procedure is the best one based on the results of this MS

Reviewer #2: Title: Comparison of shape quantification methods for genomic prediction, and genome-wide association study of sorghum seed morphology

Sorghum seed shape and size directly impact its grain yield. As seed shape cannot be reliably quantified, previous sorghum research used 100-seed mass from common garden experiments to understand the genetic basis of seed shape/size. In this study, Sakamoto et al., used two shape quantification methods [elliptic Fourier analysis (EF) and semi-landmark analysis (SLA)], previously used in rice and wheat, to quantify sorghum seed shape. This data was used to determine the genetic architecture of sorghum seed shape. The authors quantified seed shape of 329 diverse sorghum accessions using EF and SLA methods and genotyped these accessions by sequencing (GBS). Principal component analysis of EF and SLA analysis revealed both methods produced similar results. GWAS with seed shape data identified a significant peak on chromosome 4, validating this QTL was a seed weight QTL mapped recently by Wang et al., (2019, https://www.nature.com/articles/s41437-019-0249-4).

These results are interesting and potentially important for crop improvement. However, there are some minor concerns.

1. The authors used ~329 sorghum diverse accessions in this study. However, geo-reference and their race information are not provided in the manuscript. Including this information as a supplemental file will be helpful for future studies with different traits from this population. Also, include a PCA plot of these accessions plotted with the sorghum association panel (Morris et al. 2013. PNAS), and NJ tree depicting their origin and race. These figures will be helpful for sorghum genetics and genomics audience to understand the genetic diversity present in the sorghum germplasm used for this research.

2. Recently, sorghum seed mass (with ~1900 geo-referenced accessions) mapped a QTL on chromosome 4 at ~58 Mb (Wang et al., 2019). As findings from Wang et al. (2019) validate the seed shape QTL mapped in this study, cite Wang et al. in the discussion section.

Minor points:

Line 87: change Sorghum × almum to Sorghum x almum

6. PLOS authors have the option to publish the peer review history of their article (what does this mean?). If published, this will include your full peer review and any attached files.

Reviewer #1: No

Reviewer #2: No

---

## [Author Response · Author response to Decision Letter 0]

5 Oct 2019

Response to Reviewers

Reviewer #1

In the abstract, authors highlighted the methods, it would be better to give more background, and while the detailed comparison is important in introduction.

>We added the detailed comparison of geometric morphometrics in the abstract (L25-L28, L36-39).

Is the genome sequencing data (RAD-Seq data) generated in the present study? If yes, I think authors need to provide more detailer results. If not, please cite where the data from. Because I did not find any germplasm information in the MS.

> We added supporting information (S1 Table at L94 and S1 Fig at L112) for providing cultivar names, country names, a neighbor-joining tree and a PCA plot based on the SNP genotypes of these accessions.

In row 158, 168 and 190, for "function ef2cood in shape on R", authors cited two different papers. need to be corrected.

> We corrected them to cite the same paper (the reference #31).

In row 225, authors used rrBLUP to run GWAS, the paper need to be cited. And authors used Q and K in the GWAS, while not mention where the Q and K from.

>We cited a paper related to rrBULP (L263, the refenrence #34), and added methods for calculating Q and K (L240-241).

In the table 1, authors need to show more information, such as p value for the association.

> We added supporting information (S3 Table) for providing p-values of all SNPs analyzed in GWAS. 

In row 375, authors wrote "The best procedure for the genetic analysis of biological shape should be chosen based on the shape features", can authors point out which procedure is the best one based on the results of this MS

> Based on the results of this MS, we can mentioned as ”the direction and scaling standardization procedures, however, did affect the results.” (L388). However, because we cannot have a universal definition for “ the best”, it is difficult to choose “the best procedure” even in our case of dataset. To avoid the confusion, we removed the sentence. 

Reviewer #2

The authors used ~329 sorghum diverse accessions in this study. However, geo-reference and their race information are not provided in the manuscript. Including this information as a supplemental file will be helpful for future studies with different traits from this population. Also, include a PCA plot of these accessions plotted with the sorghum association panel (Morris et al. 2013. PNAS), and NJ tree depicting their origin and race. These figures will be helpful for sorghum genetics and genomics audience to understand the genetic diversity present in the sorghum germplasm used for this research.

> We added supporting information (S1 Table at L94 and S1 Fig at L112) for providing cultivar names, country names, a neighbor-joining trees and PCA plot based on the SNP genotypes of these accessions.

Recently, sorghum seed mass (with ~1900 geo-referenced accessions) mapped a QTL on chromosome 4 at ~58 Mb (Wang et al., 2019). As findings from Wang et al. (2019) validate the seed shape QTL mapped in this study, cite Wang et al. in the discussion section.

>Thank you for letting us know the paper. We cited Wang et al. (2019) in the introduction and discussion section (L61 and L384).

 Minor points: Line 87: change Sorghum × almum to Sorghum x almum

>We corrected Sorghum × almum to Sorghum x almum (L95).

---

## [Editor Report · Decision Letter 1]

21 Oct 2019

Comparison of shape quantification methods for genomic prediction, and genome-wide association study of sorghum seed morphology

PONE-D-19-15031R1

Dear Dr. Iwata,

We are pleased to inform you that your manuscript has been judged scientifically suitable for publication and will be formally accepted for publication once it complies with all outstanding technical requirements.

With kind regards,

Sujan Mamidi, Ph.D.

Academic Editor

PLOS ONE
---

## [Editor Report · Acceptance letter]

11 Nov 2019

PONE-D-19-15031R1 

Comparison of shape quantification methods for genomic prediction, and genome-wide association study of sorghum seed morphology 

Dear Dr. Iwata:

I am pleased to inform you that your manuscript has been deemed suitable for publication in PLOS ONE. Congratulations! Your manuscript is now with our production department. 

With kind regards,

on behalf of

Dr. Sujan Mamidi 

Academic Editor

PLOS ONE